# Coronavirus-Related Health Literacy: A Cross-Sectional Study during the COVID-19 Pandemic in Italy

**DOI:** 10.3390/ijerph19073807

**Published:** 2022-03-23

**Authors:** Aldo Rosano, Chiara Lorini, Brigid Unim, Robert Griebler, Chiara Cadeddu, Luca Regazzi, Daniela Galeone, Luigi Palmieri

**Affiliations:** 1National Institute for the Analysis of Public Policy, Corso d’Italia 33, 00198 Rome, Italy; a.rosano@inapp.org; 2Department of Health Sciences, University of Florence, Viale GB Morgagni 48, 50134 Florence, Italy; chiara.lorini@unifi.it; 3Department of Cardiovascular, Endocrine-Metabolic Diseases and Aging, Italian National Institute of Health, Via Giano della Bella 34, 00162 Rome, Italy; brigid.unim@iss.it; 4Competence Centre for Health Promotion and Health System, Austrian National Public Health Institute, Stubenring 6, A-1010 Vienna, Austria; robert.griebler@goeg.at; 5Department of Life Sciences and Public Health, Università Cattolica del Sacro Cuore, Largo Francesco Vito 1, 00168 Rome, Italy; chiara.cadeddu@unicatt.it (C.C.); luca.regazzi01@icatt.it (L.R.); 6Ministry of Health, Viale Giorgio Ribotta, 5, 00144 Rome, Italy; d.galeone@sanita.it

**Keywords:** health literacy, COVID-19, Italy, risk perception, knowledge, attitudes and practices, risk factors

## Abstract

The COVID-19 pandemic has caused an overabundance of valid and invalid information to spread rapidly via traditional media as well as by internet and digital communication. Health literacy (HL) is the ability to access, understand, appraise, and apply health information, making it fundamental for finding, interpreting, and correctly using COVID-19 information. A cross-sectional study of a sample of 3500 participants representative of the Italian adult population aged 18+ years was conducted in Italy in 2021. A validated HL questionnaire was employed, including sections on coronavirus-related HL, general HL, sociodemographic characteristics, risk factors, and respondents’ lifestyle. Of our sample, 49.3% had “excellent” levels of coronavirus-related HL and 50.7% had “sufficient” (20.7%) or “limited” (30.0%) levels. Although the overall HL-COVID level was high, many participants reported difficulties dealing with COVID-19 information; in particular, participants older than 65 years, with a low education level, living in southern regions of Italy, and with high financial deprivation. Targeted public information campaigns and the promotion of HL are required for better navigation of health information environments. The COVID-19 pandemic has highlighted the need to improve HL and to prepare the general population for future emergency and non-emergency situations, confirming that HL can be considered a social vaccine.

## 1. Introduction

Severe acute respiratory syndrome coronavirus 2 (SARS-CoV-2) is the cause of the coronavirus disease (COVID-19) pandemic that has been affecting the world since its first outbreak in the city of Wuhan, China, in December 2019 [1,2]. By 15 February 2022, approximately 411 million COVID-19 cases had been confirmed with more than 5.8 million deaths worldwide, and more than 165 million cases and 1.8 million deaths in Europe [3]. In Italy, the enormous impact of the COVID-19 pandemic resulted in approximately 11.8 million cases and 150 thousand deaths by 15 February 2022 [4]. Public health preparedness is a significant factor in the success of reducing COVID-19 transmission. Lessons learned from countries across the world imply the need for community-oriented strategies and rapid response from public health officials to successfully contain the disease (e.g., Eastern Asia) [5]. Such strategies imply health education and health literacy strengthening in addition to monitoring, control, and behavioral measures, such as early case identification, widespread laboratory testing and screening, outbreak mitigation (up to and including lockdowns), contact tracing, physical distancing, and quarantine measures which have been demonstrated as essential interventions to curb the pandemic [5].

The COVID-19 pandemic has caused an overflow of reliable and less reliable information to spread rapidly via traditional media (television, radio, and newspaper) as well as by internet and digital communication. Along with the COVID-19 outbreak, the circulation of mis- and disinformation on SARS-CoV-2 and COVID-19 has hindered the containment measures and become an additional threat. The deluge of both accurate and inaccurate health information has been defined as “infodemic” and constitutes a major challenge to effective health communication. Since the beginning of the pandemic, the World Health Organization (WHO) has highlighted the risk that an infodemic can prejudice public health political decisions and, consequently, preventive actions and interventions. The WHO underlined the unprecedented role that risk communication and community engagement plays in breaking the chains of transmission and reducing the impact of the COVID-19 pandemic [6]; this objective implies and requires a coordinated effort to manage the infodemic by building and improving health literacy (HL) in the population [6]. HL, in general, encompasses people’s knowledge, motivation, and competencies to find, understand, appraise, and apply health information, as well as to make judgments and informed decisions concerning healthcare, disease prevention, and health promotion, and to act accordingly [7]. Thus, it is crucial to properly navigate the coronavirus and COVID-19 information environments. 

Since 2018, Italy has participated in the Action Network on Measuring Population and Organizational Health Literacy (M-POHL) under the umbrella of the WHO European Health Information Initiative, in alignment with Health 2020, the European policy framework for health and well-being. Under the M-POHL network, 17 countries in the WHO European Region performed a coordinated survey on HL. The Italian version of the Health Literacy Population Survey 2019–2021 (HLS19) [8] was coordinated by the Italian National Institute of Health with the support of the Italian Ministry of Health. The present study investigates how Italian adults handle coronavirus- and COVID-19-specific information and examined the relationship of these information management skills with other HL dimensions making use of the information derived from specific sections of the questionnaire administered to a representative sample of Italian adults during the COVID-19 pandemic. The main objective of this study was to assess the coronavirus-related HL (HL-COVID) of Italian adults and to examine the differences in HL in relation to sociodemographic conditions.

## 2. Materials and Methods

The HLS19 [8] was implemented in Italy after a pilot phase was conducted between 31 March and 7 April 2021. From 8 April to 8 May 2021, a questionnaire including a 47-item section (HLS19-Q47)on general HL (GEN-HL), including 4 vaccination-related items (HLS19-VAC) selected to assess the vaccination-specific HL (HL-VAC), a 16-item section on HL-COVID (Corona Health Literacy Questionnaire) [9], and a 34-item section on sociodemographic characteristics was administered to a sample of 3500 men and women aged 18 years and older, representative of the Italian general population; an online (2949 interviews) and telephone (551 interviews) survey was used. The telephone survey was conducted mainly among the elderly (65 years and older) to facilitate the participation of people who might encounter major difficulties with internet use. Participants were first stratified by region and population density (urban/rural), then by sex and age. They were sampled randomly and proportionally from each stratum. When a selected participant declined to participate, a substitute from the same stratum was randomly sampled. The target population was the resident Italian population aged 18 years and older living in private households. The response rate, among those who agreed to participate after an e-mail invitation, was 58%. 

The HL-COVID section of the questionnaire included 4 thematic and 4 process domains investigating the following 8 subdimensions: Thematic subdimensions: (a) protecting oneself against infection; (b) preventing others from being infected; (c) recognizing a possible infection; and (d) dealing with a possible infection;Process subdimensions: (e) finding; (f) understanding; (g) evaluating; and (h) applying coronavirus-related information.

For all the HL-items, a 4-point Likert scale was applied: very easy, easy, difficult, and very difficult. The main HL-COVID score was based on all 16 items; the scores for the 8 subdimensions were calculated based on a subset of 4 items each. Various scoring systems concerning HL were suggested in the scientific literature, according to various tools used to investigate HL levels: Sørensen et al. [10] used a mean-based item raw score, transformed into a unified metric ranging from 0 to 50, and proposed 3 cutoff values resulting in 4 levels of HL; Röthlin et al. [11] adopted a score based on dichotomized answers and suggested merging the 2 highest HL levels; Okan et al. [12] used a mean score, 2 cutoff values, and 3 levels; the HLS19 consortium (2021) [8] chose to calculate a score based on dichotomized answers and proposed 4 cutoff levels which combine different profiles of answers (Table 1).

In the present study, we adopted an approach based on the ordinal measurement scale [13,14] and used a score based on dichotomized answers according to the HLS19 methodology. In detail, scores were calculated as the percentage of items (ranging from 0 to 100) with valid responses that were answered “very easy” or “easy”, provided that at least 80% of the items contained valid responses. If less than 80% of the items contained valid responses, the score was set to “missing”. A higher score corresponds to a higher level of HL. 

In light of the methods reported in Table 1, and considering the characteristics of the questionnaire, levels of HL-COVID were assigned to respondents according to the following definitions of the cutoff points:Excellent: “very easy” + “easy” > 81.3% (more than 12 of 16 answers);Sufficient: 50.0% < “very easy” + “easy” < 81.3% (from 9 to 12 of 16 answers);Limited: “very easy” + “easy” ≤ 50.0% (fewer than 9 of 16 answers);

And for the 8 subdimensions, according to the following definitions:Excellent: “very easy” + “easy” > 81.3% (4 of 4 answers);Sufficient: 50% < “very easy” + “easy” < 81.3% (3 of 4 answers);Limited: “very easy” + “easy” ≤ 50% (fewer than 3 of 4 answers).

The same criterion was adopted to assign the level of the 4-item HL-VAC. The internal consistency of the scores was examined by calculating Cronbach’s alpha and the Spearman-Brown split-half reliability coefficient (rhoSB). The relationship between the HL-COVID levels and the sociodemographic characteristics was investigated through the distribution of the HL-COVID score by age groups, sex, educational level, level of financial deprivation, geographic area, GEN-HL, and HL-VAC, as well as through an ordinal logistic regression (OLR) in which the sociodemographic factors were simultaneously analyzed in association with the HL-COVID levels. Education was surveyed using the International Standard Classification of Education (ISCED) 2011 scale. The ISCED codes were then recoded to 3 levels: “low”, corresponding with lower secondary education or below (up to ISCED-2); “medium”, corresponding with higher secondary education (ISCED-3); “high”, corresponding with postsecondary or short-cycle tertiary education (ISCED-4 and 5), or with bachelor or higher (ISCED-6 to 8). Financial deprivation was defined according to 3 specific items investigating the capacity to afford medication, medical examinations and treatments, and to pay all bills at the end of the month. The financial deprivation scores were calculated as percentages (ranging from 0 to 100) for items with valid responses that were answered with “very difficult” or “difficult”. The values of the financial deprivation score were assigned the following level: 0% = none, 33.33% = some, 66.66% = considerable, 100% = severe. The regression analysis was conducted adopting a proportional odds model, i.e., the effects of all explanatory variables are proportional to the different threshold values of the outcome variable (e.g., the HL-COVID level). The suitability of a proportional odds logistic regression model depends on the assumption that each input variable has a similar effect on the different levels of the ordinal outcome variable. An odds ratio higher than 1 identify the situation of increased risk of lower levels of HL. The association of the HL-COVID score with the GEN-HL score (the general score of the overall HL “core” questionnaire based on 47 items) and with HL-VAC was investigated through a correlation analysis.

## 3. Results

Internal consistency of the HL-COVID measure was very high (alpha = 0.96; rhoSB = 0.95). The eight subscales also showed high internal consistency (subscale “protect”: alpha = 0.85; “prevent”: alpha = 0.89; “recognize”: alpha = 0.84; “dealing”: alpha = 0.89; “finding”: alpha = 0.84; “understanding”: alpha = 0.85; “evaluating”: alpha = 0.86; “applying”: alpha = 0.83). The main HL-COVID score was 68.5. Scores (percentage of valid answers: easy + very easy) of the subdimensions were as follows: protecting against infection (71.9); preventing other people from being infected (71.5); recognizing a possible infection (60.7); dealing with infection (66.3); finding information (74.6); understanding information (73.8); evaluating coronavirus-related information (59.1); and applying the correct information (66.0). Considering the perceived difficulties at the item level, the percentage of valid answers “difficult + very difficult” ranged between 19.0% (CHL1) and 42.7% (CHL8). Items CHL7 (42.7%) and CHL3 (42.19%) were rated by the respondents as extremely difficult tasks (Table 2).

Of our sample, 49.3% had excellent levels of HL-COVID, and 50.7% had “sufficient” (20.7%) or “limited” (30.0%) levels. Although the majority of interviewed persons had a good level of HL-COVID, many participants reported difficulties dealing with coronavirus- and COVID-19-related information. In particular, the individuals with limited HL-COVID levels were those older than 65 years (32.5%), with low education levels (31.8%), living in the ‘South and islands’ geographical area (32.3%), and with severe financial deprivation (46.4%). There were no noteworthy sex differences in HL-COVID levels (Table 3). The HL-COVID score was positively correlated with the GEN-HL score (r = 0.60) as well as with HL-VAC (r = 0.51). Among those with low GEN-HL (first quartile), 12.4% had excellent levels of HL-COVID; for those with GEN-HL in the second quartile, 36.0% had levels of excellent HL-COVID; for those with GEN-HL in the third quartile, 60.9% had excellent levels of HL-COVID; and for those with GEN-HL in the fourth quartile, 88.5% had excellent levels of HL-COVID. Those with high levels of HL-COVID also had high levels of HL-VAC, and only a small percentage (17.8%) of those with excellent HL-COVID levels had a limited level of HL-VAC (Table 3).

In Table 4 and Table 5, the HL-COVID scores (percentage of valid answers: easy + very easy) of the eight subdimensions “protect”, “prevent”, “recognize”, “dealing” (Table 4), and “find”, “understand”, “evaluate”, and “apply” (Table 5) are reported by the level of coronavirus-related HL (excellent, sufficient, limited) and by participant characteristic (sex, age class, educational level, geographic area, deprivation level, quartile of GEN-HL score based on the general questionnaire of 47 items, and HL-VAC).

As shown in Table 4, individuals with “excellent” HL-COVID in the four thematic subsections were mostly in the 18–29 age group and scored between 39.7% (“recognize”) and 63.2% (“prevent”). Excellent HL-COVID was also observed in respondents with a medium educational level (38.1% in the subsection “recognize”) and in those with a high educational level (from 51.4% “deal” to 58.0% “prevent”). Likewise, HL-COVID was “excellent” in the respondents living in central Italy (from 39.2% “recognize” to 58% “prevent”), in those with GEN-HL in the fourth quartile (from 69.4% “recognize” to 88.3% “prevent”) and among those with excellent HL-VAC (from 56.0% “recognize” to 75.5% “prevent”). In all four subdimensions, the highest prevalence of “excellent” HL-COVID was registered in the “no deprivation” group; for the other deprivation conditions, excellent HL-COVID was mostly observed in those with “some” financial deprivation, who scored from 35.6% “recognize” to 53.2% “prevent”.

As reported in Table 5, for both sexes, the majority of respondents had an “excellent” level of HL-COVID for the subdimensions “find” and “understand,” with a slight predominance in women compared with men, whereas both sexes were predominantly “limited” in the subdimensions “evaluate” and “apply”. In all the subdimensions included in the table, with some differences in the subdimension “apply,” prevalence of the “excellent” level of HL-COVID increased with decreasing age class, which was the opposite in those at the “limited” HL-COVID level. The prevalence of an “excellent” level of HL-COVID increases with increasing educational level in all the subdimensions; the opposite was registered in the “limited” HL-COVID level. No geographical gradient appeared to be evident in all four subdimensions for the “excellent” level of HL-COVID; the highest prevalence of the “excellent” level is registered in the central geographical area for all the subdimensions. The highest prevalence of the “limited” HL-COVID level was in the south and islands geographical area for the subdimensions “find”, “understand”, and “apply”, and in the northeast area for the subdimension “evaluate”. In all the four considered subdimensions, the prevalence of an “excellent” level of HL-COVID decreased with the increase in the deprivation level; the opposite was found in the “limited” HL-COVID level. In the subdimensions “find” and “understand”, the absolute majority were persons with an “excellent” level of HL-COVID and no deprivation. The prevalence of an “excellent” level of HL-COVID increased with the increase in GEN-HL quartile based on the overall questionnaire of 47 items in all the four subdimensions as well as with the increase in HL-VAC level; the opposite was found in the “limited” HL-COVID level. The majority of the “excellent” level was registered in the fourth quartile of the GEN-HL level in all four subdimensions.

The results of the OLR analysis are shown in Figure 1. Odds ratios (ORs) and 95% confidence intervals of the sociodemographic factors were reported in relation to the HL-COVID levels. Considering the sociodemographic factors simultaneously in the regression model, only the age classes older than 45 years and the financial deprivation classes were significantly associated with HL-COVID levels. In particular, the odds of sufficient/limited HL-COVID levels increased by 39% in persons aged from 45 to 64 years and by 67% in persons aged 65 years and older compared with persons aged 18–29 years (reference category for age). Similarly, the odds of a low HL-COVID level (sufficient/limited) increased by 50%, 92%, and almost triple in persons with a low, considerable, and severe financial deprivation level, respectively.

## 4. Discussion

The COVID-19 pandemic has a strong impact on people’s lives, involving both the practical aspects, such as the effects of applying preventive and protective behaviors and the emotional impacts of the epidemiological situation and the restrictions. However, the effectiveness of the pandemic control measures requires a broad understanding from the population on the importance of individual behaviors in protecting and promoting population health. In this respect, analyzing HL in the context of COVID-19 is fundamental to gain insights into how people feel informed about the coronavirus or confused by the sheer amount of coronavirus- and COVID-19-related information.

The present study assessed the HL-COVID level of an Italian representative sample during the COVID-19 pandemic. Half of the participants were excellently capable of dealing with coronavirus- and COVID-19-related information, specifically when it came to finding and understanding information, protecting themselves from infection, and preventing others from becoming infected. The easiest task was to “find or obtain information on how to protect yourself against infection with the coronavirus.” However, 30% reported limited levels of HL-COVID, with “dealing with information on coronavirus infection” being the most problematic issue. The most difficult tasks were “to judge whether the information on coronavirus symptoms indicating a possible coronavirus infection that can help to decide if you may be infected”, was reliable, and whether information on how “to protect yourself from infection” was reliable. Similar difficulties were found in the coronavirus-related HL survey performed in Germany [12], as well as in other studies conducted in various target groups [15]. These results highlight a specific lack of information that need to be addressed with tailored intervention, although it is important to note that the judgment on information about coronavirus symptoms indicating they had a possible coronavirus infection could be influenced by the emotional distress related to the fear of being infected, in addition to skills and competences. On the other hand, the correlation between HL-COVID, GEN-HL, and HL-VAC suggests that general and specific skills are related, and the dimensions are, at least partly, overlapping. The European Health Literacy Survey, which was conducted in eight countries (Austria, Bulgaria, Germany, Greece, Ireland, Poland, Spain, and the Netherlands), found that almost 50% of all adults had “problematic” or “inadequate” GEN-HL, meaning that it is potentially difficult for them to access, understand, appraise, and apply information to promote or protect their health [10]. The HLS19 survey, conducted in 17 European countries, found that 33% of respondents have a “problematic” level of HL and 13% an “inadequate” level [8]. This issue requires targeted public information campaigns and the promotion of HL for better navigation of information environments during the current infodemic regarding COVID-19, as well as for other problematic public health issues.

Previous studies have demonstrated that low HL is associated with various adverse health outcomes [10,16], higher healthcare costs [17,18,19], and unhealthy behaviors [20]. In addition, several studies have demonstrated an association between low HL and financial deprivation, educational level, social status, age, and sex [10,21,22,23]. In the current study, difficulties with coronavirus- and COVID-19-related information were encountered by those who had lower GEN-HL, severe financial deprivation, and were 65 years of age or older; confirming that their difficulties were the result of social determinants. However, differences in sex and education had only a minor effect on the HL-COVID score in our study, which is similar to a previous study conducted in an Italian city [24]. Okan et al. [12], contrary to our findings, did not report significant differences in HL-COVID related to age and income in Germany. These findings indicate that HL varies considerably across countries and that various subgroups of the population in each country are impacted to a different degree [25,26]. This variation constitutes an important public health challenge that should be swiftly addressed by developing appropriate public health strategies. On the other hand, the correlation between GEN-HL and HL-COVID suggests the need to invest in promoting the general HL of the population to make people more able to forecast events requiring specific skills in the future, as has occurred with the COVID-19 pandemic.

In a study conducted in the region of Tuscany, in Central Italy, general HL was found to be a predictor of both better self-reported knowledge and an attentive attitude toward the importance of COVID-19 preventive measures, whereas it had no role in predicting a higher risk perception [15]. Similar results have also been observed in a Japanese survey [27]. In the present study, high levels of HL-COVID are associated with excellent HL-VAC, especially in the subdimensions ‘find’, ‘understand’, and ‘prevent’. These findings underline a relationship between knowledge and confidence in the safety and effectiveness of vaccines, which in turn are correlated with vaccination coverage rates. Tailored vaccination campaigns and improving HL among the general population could be the key solution to vaccine hesitancy. Understanding public health recommendations, applying protective measures against infection with coronavirus, and navigating COVID-19-related health information environments are currently of elevated importance [28,29], which underlines the need to explore HL-COVID.

In accordance with our findings, a recent survey conducted in Switzerland reported that a large number of respondents had difficulties assessing the trustworthiness of information from the media [30]. The lack of citizens’ trust in the media inevitably raises doubts about national and evidence-based public health strategies, leading to refusals to adhere to recommended actions. Thus, there is a need for policy and public health measures to counter the coronavirus infodemic and increase the trust of the general population in national health plans and recommendations.

This study has some limitations. Although the sampling procedure reproduced the overall Italian adult population in terms of age, sex, and geographic distribution, the survey methods used for the interviews (internet and landline phone) might exclude those without internet access or a landline phone. For this reason, the generalizability of the results to the entire Italian population could be limited. As with all web-based surveys, the sample composition and self-selection bias constitute other issues limiting the value of the findings [31]. Additionally, the results could have been influenced by a nonresponse bias. Moreover, given that a subjective tool was used to measure HL, the findings could be affected by overestimation or underestimation of perceived abilities by individual participants [24]. In particular, what people think they know does not always correspond to what they actually know: people tend to be overconfident (they think they know more than they actually do) or underconfident (they think they know less than they actually do). Overconfidence and underconfidence are a consequence of the matching between knowledge, confidence, self-efficacy, and emotional distress, and may differ from country to country, as they are also influenced by cultural factors. Furthermore, overconfidence and underconfidence can also be affected by individual factors, such as age, educational level, empowerment, and self-efficacy [24,32,33].

## 5. Conclusions

The COVID-19 pandemic has highlighted the need to improve HL and prepare the general population for future emergency and non-emergency situations, confirming that HL can be considered a social vaccine—an asset to empower citizens to protect their own health and the health of the population—with a stronger effect on vulnerable and high-risk groups. Moreover, according to our results, enhancing general HL could also affect the improvement of specific skills, such as those related to COVID-19 or to vaccination. Comparative studies considering different populations in different pandemic phases would be useful to better investigate the role of HL with respect to knowledge and behaviors from the public health perspective, as well as to identify target groups for intervention. In particular, focusing on adolescents and younger people could help understand how to better control the spread of COVID-19.

## Figures and Tables

**Figure 1 ijerph-19-03807-f001:**
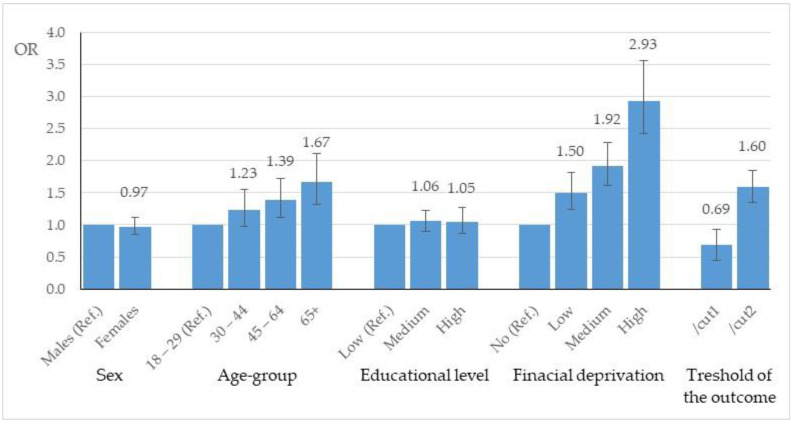
Association between HL-COVID main score level and participant characteristics (sex, age, education, financial deprivation) in terms of odds ratio (OR) with 95% confidence interval.

**Table 1 ijerph-19-03807-t001:** Health literacy instruments, scores, and cutoff levels.

Instrument	*N*. of Items	Score	Transformation	Cutoff Values
HLS-EU-Q47 [10]	47	Mean	Scores transformed to a unified metric with a minimum of 0 and a maximum of 50, where 0 represents the ‘least possible’ and 50 represents the ‘best possible’ health literacy score.	0–25 = “inadequate”; >25–33 = “problematic”; >33–42 = “sufficient”; >42–50 = “excellent”
HLS-EU-Q16 [11]	16	Sum	The answers were dichotomized: “very difficult/rather difficult” versus “easy/very easy”	More than 12 out of 16 answers = “Sufficient”; From 9 to 12 out of 16 answers = “Problematic”; Less than 9 out of 16 answers = “Inadequate”
HLS-COVID-Q22 [12]	22	Mean	No	≤2.5 = “inadequate”; >2.5 and <3 = “problematic”; ≥3 = “sufficient”
HLS19-Q12 [8]	12	Sum	The answers were dichotomized: “very difficult/difficult” versus “easy/very easy”	If the number of answers “very easy” is above 1/2 and the answers for “(very) difficult” is not more than 1/12 = “excellent”; if at least 10 out of 12 items answers are “(very) easy” and not more than 2 out of 12 are “(very) difficult” = “sufficient”; if the number of answers “(very) difficult” is above 1/2 and for “very easy” is not more than 1/12 = “Inadequate”; the intersecting set of not “excellent” and not “sufficient” and not “inadequate” = “problematic”

**Table 2 ijerph-19-03807-t002:** Percentages of respondents by category and type of question of the “Corona Health Literacy Questionnaire”.

Code	On a Scale From “Very Easy” to “Very Difficult”, How Easy or Difficult Would You Say It Is:	Very Difficult	Difficult	Easy	Very Easy	Subdimensions
CHL1	to find/obtain information on how to protect yourself against infection with the coronavirus?	3.5	15.5	56.9	24.1	Protect/Find
CHL2	to understand information on how to protect yourself from infection with the coronavirus?	3.4	18.3	56	22.24	Protect/Understand
CHL3	to judge if the information on how to protect yourself from infection with the coronavirus is reliable?	9.1	33.1	45	12.84	Protect/Evaluate
CHL4	to base your own decisions on information on how to protect yourself from infection with the coronavirus?	4.6	24.7	52.8	17.93	Protect/Apply
CHL5	to find/obtain information about symptoms indicating a possible coronavirus infection that can help you decide if you may be infected?	4.1	24.1	54.1	17.7	Recognize/Find
CHL6	to understand information about symptoms indicating a possible coronavirus infection that can help you decide if you may be infected?	4.8	25.4	52.5	17.31	Recognize/Understand
CHL7	to judge if information about symptoms indicating a possible coronavirus infection that can help you decide if you may be infected is reliable?	8.5	34.2	44.7	12.62	Recognize/Evaluate
CHL8	to decide if you are possibly infected based on information about symptoms of a possible coronavirus infection?	7.4	35.4	44.8	12.48	Recognize/Apply
CHL9	to find/obtain information about what to do if you think you are infected with the coronavirus?	5.8	24.9	52.8	16.54	Deal/Find
CHL10	to understand information about what to do if you think you are infected with the coronavirus?	5.7	23.8	53.8	16.75	Deal/Understand
CHL11	to judge if information about what to do if you think you are infected with the coronavirus is reliable?	8.1	32.8	44.9	14.16	Deal/Evaluate
CHL12	to base your own decisions on information about what to do if you think you are infected with the coronavirus?	5.6	28.0	51.0	15.38	Deal/Apply
CHL13	to find/obtain information about what you can do to prevent other people from being infected with the coronavirus?	4.1	19.5	58.2	18.17	Prevent/Find
CHL14	to understand information about what you can do to prevent other people from being infected with the coronavirus?	3.9	19.3	58.3	18.49	Prevent/Understand
CHL15	to judge if information about what you can do to prevent other people from being infected with the coronavirus is reliable?	6.9	30.1	48.8	14.27	Prevent/Evaluate
CHL16	to base your own decisions on information about what you can do to prevent other people from being infected with the coronavirus?	5.2	25.0	53.9	15.88	Prevent/Apply

**Table 3 ijerph-19-03807-t003:** HL-COVID levels by participants’ sociodemographic characteristics.

Study Population			Coronavirus-Related Health Literacy Levels (%)		
			Excellent	Sufficient	Limited
Total (*n* = 3500)					
Sex	*n*	%			
Male	1685	48.1	49.9	20.2	29.9
Female	1815	51.9	48.7	21.2	30.1
Age					
18–29	468	13.4	56.6	21.6	21.8
30–44	826	23.6	50.6	21.2	28.2
45–64	1254	35.8	49.1	18.5	32.4
65+	952	27.2	44.8	22.8	32.5
Education					
Low	1470	42.0	47.2	21.0	31.8
Medium	1299	37.1	49.7	20.6	29.7
High	731	20.9	52.8	20.3	27.0
Geographic area					
Northwest	939	26.8	50.3	19.3	30.5
Northeast	710	20.3	45.2	24.7	30.1
Center	681	19.5	54.2	20.6	25.3
South and islands	1170	33.4	48.1	19.6	32.3
Financial deprivation					
No	1478	44.6	57.7	21.0	21.2
Low	539	16.3	50.1	20.2	29.7
Medium	743	22.4	43.3	21.8	34.9
High	552	16.7	34.6	19.0	46.4
Quartile of GEN-HL					
First	896	25.6	12.4	17.9	69.8
Second	854	24.4	36.0	32.7	31.4
Third	875	25.0	60.9	25.5	13.6
Fourth	875	25.0	88.5	7.2	4.3
HL Vaccination					
Excellent	1580	47.5	70.1	17.2	12.7
Sufficient	670	20.1	36.4	27.3	36.4
Limited	1077	32.4	17.8	20.1	62.1

**Table 4 ijerph-19-03807-t004:** HL-COVID levels of thematic subdimensions by participant characteristics.

Study Population	Protect			Prevent			Recognize			Deal		
	Excellent	Sufficient	Limited	Excellent	Sufficient	Limited	Excellent	Sufficient	Limited	Excellent	Sufficient	Limited
Total (*n* = 3500)												
Sex												
Male	49.5	18.1	32.4	53.1	15.4	31.5	35.1	15.4	45.4	47.1	15.6	37.3
Female	48.1	19.8	32.1	54.6	13.7	31.6	36.6	13.7	44.4	46.6	15.5	38.0
Age class												
18–29	52.2	21.8	26.0	63.2	13.4	23.4	39.7	20.1	40.2	56.6	15.2	28.2
30–44	46.6	17.6	35.9	55.4	14.0	30.6	33.2	21.4	45.4	48.5	16.8	34.7
45–64	50.5	17.6	32.0	53.2	13.7	33.1	38.0	18.4	43.7	46.2	14.1	39.7
65+	46.8	20.7	32.5	48.9	16.6	34.5	33.7	18.0	48.4	41.4	16.5	42.1
Education												
Low	45.9	19.2	34.9	50.7	15.0	34.3	33.8	19.7	46.5	44.3	16.6	39.1
Medium	50.4	18.5	31.1	55.2	14.3	30.5	38.1	18.2	43.7	47.0	14.7	38.2
High	51.7	19.4	28.9	58.0	14.0	28.0	36.2	20.0	43.8	51.4	14.9	33.7
Geographic area												
Northwest	49.0	18.1	32.9	54.6	13.1	32.3	36.4	19.0	44.5	48.4	13.8	37.8
Northeast	45.0	23.1	31.9	48.8	18.5	32.7	32.8	20.3	46.9	43.3	19.4	37.3
Center	51.0	20.8	28.1	58.0	15.4	26.6	39.2	21.2	39.6	49.6	17.2	33.2
South and islands	49.6	16.0	34.3	54.0	12.7	33.3	35.5	17.5	47.0	46.1	13.5	40.4
Deprivation												
No	56.0	20.6	23.4	61.4	15.1	23.5	40.5	22.3	37.2	54.4	16.0	29.7
Some	47.8	17.5	34.7	53.2	13.7	33.1	35.6	18.9	45.5	47.6	15.2	37.1
Considerable	43.8	18.7	37.5	49.8	14.9	35.3	32.6	17.7	49.8	40.9	15.8	43.3
Severe	37.2	16.5	46.4	40.5	14.2	45.3	27.3	15.2	57.5	32.8	15.9	51.3
GEN-HL quartile												
First	16.9	19.2	63.9	18.6	14.1	67.3	10.1	11.7	78.2	13.3	11.3	75.4
Second	39.1	26.1	34.8	44.8	20.2	35.0	25.6	20.8	53.7	33.9	22.8	43.3
Third	55.3	23.4	21.3	64.3	18.3	17.4	39.1	27.5	33.4	54.9	20.8	24.3
Fourth	84.4	7.5	8.2	88.3	5.8	5.9	69.4	17.1	13.5	85.5	7.7	6.8
HL Vaccination												
Excellent	70.7	13.9	15.4	75.5	11.0	13.5	56.0	18.6	25.4	69.3	12.6	18.1
Sufficient	39.5	27.4	33.1	47.9	21.1	31.1	25.7	25.4	48.9	38.8	20.4	40.8
Limited	23.2	21.0	55.8	26.0	15.5	58.5	14.1	16.1	69.8	18.4	16.7	64.9

**Table 5 ijerph-19-03807-t005:** HL-COVID levels of process subdimensions by participant characteristics.

Study Population	Find			Understand			Evaluate			Apply		
	Excellent	Sufficient	Limited	Excellent	Sufficient	Limited	Excellent	Sufficient	Limited	Excellent	Sufficient	Limited
Total (*n* = 3500)												
Sex												
Male	52.7	18.9	28.5	53.3	16.6	30.2	41.8	14.5	43.8	37.8	20.0	42.2
Female	55.3	16.6	28.2	55.0	16.1	29.0	39.5	13.7	46.8	38.8	20.3	41.0
Age class												
18–29	61.1	16.3	22.6	57.8	17.9	24.3	48.8	17.7	33.6	42.3	24.3	33.4
30–44	50.9	20.9	28.2	53.3	17.4	29.3	42.6	14.9	42.5	34.8	22.7	42.5
45–64	53.9	16.9	29.3	54.7	15.2	30.1	41.2	11.8	47.0	39.8	18.3	42.0
65+	53.6	16.6	29.9	52.3	16.1	31.7	34.1	14.7	51.2	37.4	18.3	44.2
Education												
Low	53.3	16.6	30.1	52.0	16.5	31.5	38.2	14.1	47.7	36.9	19.3	43.8
Medium	54.2	17.7	28.1	54.6	15.5	29.9	41.9	13.6	44.6	39.2	20.5	40.3
High	55.1	19.7	25.2	57.6	17.5	24.9	43.2	14.9	41.9	39.6	21.1	39.3
Geographic area												
Northwest	55.6	16.3	28.1	55.0	15.6	29.3	42.4	13.3	44.3	39.3	19.3	41.3
Northeast	54.4	18.8	26.8	53.7	17.4	28.9	33.8	15.8	50.4	35.6	21.2	43.1
Center	56.9	19.6	23.5	57.7	17.2	25.1	44.2	14.3	41.5	42.6	20.9	36.4
South and islands	50.9	16.8	32.2	51.7	15.5	32.7	41.2	13.5	45.3	36.6	19.6	43.8
Deprivation												
No	65.2	15.1	19.7	65.2	14.1	20.7	47.3	13.0	39.8	46.0	21.1	33.0
Some	49.3	20.1	30.6	50.3	19.8	29.9	41.9	15.0	43.1	36.8	18.6	44.6
Considerable	46.0	21.7	32.4	46.1	19.1	34.8	34.1	16.4	49.6	32.3	20.7	47.0
Severe	39.4	16.8	43.8	38.2	16.9	45.0	28.9	14.1	57.0	26.1	19.7	54.3
GEN-HL quartile												
First	19.3	17.3	63.4	17.1	16.7	66.2	10.0	8.4	81.7	10.3	15.2	74.5
Second	45.7	26.4	27.9	44.3	24.0	31.7	25.6	19.3	55.1	26.5	23.1	50.4
Third	64.3	20.8	14.9	66.4	18.8	14.8	45.7	20.8	33.5	46.9	25.2	27.9
Fourth	87.1	6.6	6.3	89.2	6.0	4.8	81.6	8.3	10.2	70.2	17.3	12.6
HL Vaccination												
Excellent	74.5	13.5	12.0	76.8	11.8	11.4	62.8	13.3	23.9	57.2	21.6	21.2
Sufficient	48.6	23.7	27.7	45.7	24.1	30.3	29.6	20.1	50.3	32.0	21.4	46.6
Limited	27.1	19.3	53.6	25.9	17.9	56.2	15.1	11.3	73.7	14.7	17.4	67.9

## Data Availability

The dataset generated and analyzed during the current study is available from the corresponding author on reasonable request, according to the Data Protection Officer of the Istituto Superiore di Sanità, Rome, Italy.

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
