# Peer review of "Coronavirus-Related Health Literacy: A Cross-Sectional Study during the COVID-19 Pandemic in Italy"

_ijerph, 2022, doi:10.3390/ijerph19073807_

Round 1
Reviewer 1 Report
The topic of the article is interesting and fits well with the Special Issue "Health literacy in the Mediterranean countries". However, it is almost a mirror work to that already published in IJERPH two years ago (Okan, O; Bollweg, TM; Berens, EM; Hurrelmann, K; Bauer, U; Schaeffer, D. Coronavirus- Related Health Literacy: A 400 cross sectional study in adults during the COVID-19 infodemic in Germany. Int. J. Environ. Res. Public Health 2020, 17, 5503). Please pay attention and modify some phrases that have apparently been taken up by the aforementioned work, especially in the introduction section, in order to avoid taking risks related to plagiarism.
The basic statistics has been well conducted. Instead of or supplementing table 5, the authors could insert a histogram that takes into account the variables and the OR (with the aim of capturing the reader's attention more).
However, there is a critical aspect. The study, conducted almost a year ago, excludes, among the items aimed at testing the level of Health Literacy, the great issue of vaccination. The main health and social problem linked to COVID-19 is in fact, currently, the reduced knowledge of vaccines as a useful tool to stem the pandemic. If obviously it is not possible to integrate these data, it is necessary to specify them perhaps by expanding the limits section of the study.
The conclusions are too poor. Please expand them by mentioning, for example, some future perspectives, the need to undertake comparative studies between the different populations of the Mediterranean, with the aspects of this particular pandemic phase, the possibility of including adolescents who at present probably represent the major vectors of contagion. These are just some of the suggestions. If the authors do not give the study a future viewpoint, it risks remaining obsolete.
Author Response
Dear Editor,
We are submitting the revised version of the manuscript ijerph-1632975 entitled “Coronavirus-Related Health Literacy: A Cross-Sectional Study during the COVID-19 Pandemic in Italy”, the revised sections are marked in yellow. We would like to thank the reviewers for their comments and efforts to improve our manuscript. The following are our replies to the Reviewers’ queries.
Reviewer 1
Comments and Suggestions for Authors
The reviewer wrote: The topic of the article is interesting and fits well with the Special Issue "Health literacy in the Mediterranean countries". However, it is almost a mirror work to that already published in IJERPH two years ago (Okan, O; Bollweg, TM; Berens, EM; Hurrelmann, K; Bauer, U; Schaeffer, D. Coronavirus- Related Health Literacy: A 400 cross sectional study in adults during the COVID-19 infodemic in Germany. Int. J. Environ. Res. Public Health 2020, 17, 5503). Please pay attention and modify some phrases that have apparently been taken up by the aforementioned work, especially in the introduction section, in order to avoid taking risks related to plagiarism.
R: Phrasing of the introduction section was revised by the authors and a mother-tongue expert revised the entire manuscript for professional language editing.
The reviewer wrote: The basic statistics has been well conducted. Instead of or supplementing table 5, the authors could insert a histogram that takes into account the variables and the OR (with the aim of capturing the reader's attention more).
R: We followed the reviewer’s suggestion and replaced table 5 with a graph.
The reviewer wrote: However, there is a critical aspect. The study, conducted almost a year ago, excludes, among the items aimed at testing the level of Health Literacy, the great issue of vaccination. The main health and social problem linked to COVID-19 is in fact, currently, the reduced knowledge of vaccines as a useful tool to stem the pandemic. If obviously it is not possible to integrate these data, it is necessary to specify them perhaps by expanding the limits section of the study.
R: We included in the analysis data of the level of HL related to vaccination, derived from four specific questions included in the general HL questionnaire (HLS19-Q47). We added the following information:
- Methods (lines 87-89): ”… a questionnaire including … including 4 vaccination-related items selected to assess the vaccination-specific HL (HL-VAC) … was administered…”; (lines 143-144) “The same criterion was adopted to assign the level of the 4-item HL-VAC”; and (lines 167-168) “The association of HL-COVID score with the GHL-Q47 score (the general score of the overall HL ‘core’ questionnaire based on 47 items) and with HL-VAC was investigated through a correlation index.”
- Results (lines 198-200): “Those with high levels of HL-COVID had also high levels of HL-VAC, and only a small percentage (17.8%) of those with excellent HL-COVID level had limited level of HL-VAC (Table 3).”; (lines 204-209) “In Table 4a and 4b, the HL-COVID scores … are reported by … participant characteristic (sex, age class, educational level, geographic area, deprivation level, quartile of the total HL score based on the general questionnaire of 47 items, and HL-VAC)”; (line 214-217) “HL-COVID was “excellent”… among those with excellent HL-VAC (from 56.0% [“recognize”] to 75.5% [“prevent”])”; (lines 240-242) and “The prevalence of an “excellent” level of coronavirus-related HL increased with… the increase of HL-VAC level;”
- Discussion (lines 292-294): “On the other hand, the correlation between HL-COVID, GHL-Q47 and HL-VAC suggests that general and specific skills are related, and the dimensions are, at least partly, over-lapped”; (lines 325-329) “In the present study, high levels of general HL are associated with excellent HL-VAC, especially in the subdimensions ‘find’, ‘understand’ and ‘prevent’. These findings underline a relationship between knowledge and confidence in the safety and effectiveness of vaccines, which in turn are correlated with vaccination coverage rates. Tailored vaccination campaigns and improving general HL among the general population could be the key solution to vaccine hesitancy”; and “Moreover, according to our results, enhancing general HL could also affect the improvement of specific skills, such as those related to COVID-19 or to vaccination”.
The reviewer wrote: The conclusions are too poor. Please expand them by mentioning, for example, some future perspectives, the need to undertake comparative studies between the different populations of the Mediterranean, with the aspects of this particular pandemic phase, the possibility of including adolescents who at present probably represent the major vectors of contagion. These are just some of the suggestions. If the authors do not give the study a future viewpoint, it risks remaining obsolete.
R: The conclusions have been enriched by including the future perspectives (lines 363-368): “Moreover, according to our results, enhancing general HL could also affect the improvement of specific skills, such as those related to COVID-19 or to vaccination. Comparative studies considering different populations in different pandemic phases will be useful to better investigate the role of HL with respect to knowledge and behaviors from the public health perspective, as well as to identify target groups for intervention. In particular, focusing on adolescents and younger people could help in understanding how to better control the spread of COVID-19.”
Reviewer 2 Report
Rosano A. and colleagues have prepared a very interesting study on a
particularly compelling argument, especially in Italy. The effect that the
infodemic can have on the understanding of different aspects related to the
SARS-CoV-2 pandemic and in particular on the ability to understand the topics.
The study is very interesting, the methodology used is appropriate, the research is correct and the conclusions are consistent with the results described.
The only point that in my opinion deserves to be added in discussion is that,
as in all studies conducted with questionnaires, there is the possibility
that the judgment given on one's ability to understand is, unconsciously,
incorrect. It is known that people unable to understand a topic may at the
same time not be able to understand their inability and therefore can report
that they understand what has been said easily; vice versa people who
understand a topic very well may report that they found it particularly
difficult to understand. I believe this should be discussed in the paper.
I have no other suggestions for the authors.
Author Response
Dear Editor,
We are submitting the revised version of the manuscript ijerph-1632975 entitled “Coronavirus-Related Health Literacy: A Cross-Sectional Study during the COVID-19 Pandemic in Italy”, the revised sections are marked in yellow. We would like to thank the reviewers for their comments and efforts to improve our manuscript. The following are our replies to the Reviewers’ queries.
Reviewer 2
The reviewer wrote: Rosano A. and colleagues have prepared a very interesting study on a particularly compelling argument, especially in Italy. The effect that the infodemic can have on the understanding of different aspects related to the SARS-CoV-2 pandemic and in particular on the ability to understand the topics.
The study is very interesting, the methodology used is appropriate, the research is correct and the conclusions are consistent with the results described.
The only point that in my opinion deserves to be added in discussion is that, as in all studies conducted with questionnaires, there is the possibility that the judgment given on one's ability to understand is, unconsciously, incorrect. It is known that people unable to understand a topic may at the same time not be able to understand their inability and therefore can report that they understand what has been said easily; vice versa people who understand a topic very well may report that they found it particularly difficult to understand. I believe this should be discussed in the paper.
R: The discussion was integrated mentioning the issue of “overconfidence/underconfidence” when filling a questionnaire (lines 350-357): “In particular, what people think they know does not always correspond to what they actually know: people tend to be overconfident (they think they know more than they actually do) or underconfident (they think they know less than they actually do). Overconfidence and underconfidence are a consequence of the matching between knowledge, confidence, self-efficacy and emotional distress, and they may differ from country to country, as they are also influenced by cultural factors. Furthermore, overconfidence and under-confidence can also be affected by individual factors, such as age, educational level, empowerment and self-efficacy”
Reviewer 3 Report
The manuscript presents a relevant topic given the current health context. It is interesting to note the guidance provided on possible future measures to be implemented to increase literacy in the population. I also congratulate you on the effort you have made to obtain a fairly representative sample of the population. However, as in any research study, some details need to be reviewed:
- Lines 81-83: Please consider transforming this sentence as the main objective of this study ("The main objective of this study is to assess...").
- Table 1: There is no need to comment on the more specific aspects of HLS-EU-Q16 (for example) if it is not mentioned in the rest of the manuscript.
- Line 112: Consider changing "Sørensen and colleagues" to "Sørensen et al." to improve the uniformity of wording in the manuscript when referring to multiple authors.
- Lines 112, 148-152 and 390: I think there are double spaces and extra spaces in these lines.
- Lines 157, 167 and 193-194: Similar to the previous recommendation, there are spaces between signs and numbers (when on other occasions, e.g. on the same line 157, there are no such spaces). Please revise this detail throughout the manuscript to improve the uniformity of the presentation of numerical results.
- Lines 124-127: Did the online questionnaire system allow for the mandatory answering of items to avoid missings? After reading these lines, it seems that it was not possible.
- Line 182: Please add the name of the questionnaire.
- Table 2: Could you transcribe the questionnaire items/questions exactly? Grammatically, they are not well written, and this would allow the study to be reproduced in other geographical and temporal contexts. Depending on the rewrite, you may need to change the heading "Question" to another label.
- Line 189: Southern or Southwest? On the other hand, you may consider "South and Islands", as this is the term used to refer to this geographical area. Also, if you are going to make any changes, please note that this information also appears in the abstract.
- Lines 218 and 328, Table 5: Please change "gender" to "sex" (or "genders" to "sexes"). The sex variable is biological, which is the variable used in this study. More information is available in the section "Sex and Gender in Research" (https://www.mdpi.com/journal/ijerph/instructions#ethics).
- Line 312: Add the geographical location of Tuscany in Italy, to be able to compare with the geographical divisions you have established for this study.
- Lines 338-342: The need to increase health literacy on coronavirus in terms of recognition of infection and symptoms should be emphasised, as these have been the most difficult tasks for respondents.
I hope you find my recommendations helpful.
Best regards.
Author Response
Dear Editor,
We are submitting the revised version of the manuscript ijerph-1632975 entitled “Coronavirus-Related Health Literacy: A Cross-Sectional Study during the COVID-19 Pandemic in Italy”, the revised sections are marked in yellow. We would like to thank the reviewers for their comments and efforts to improve our manuscript. The following are our replies to the Reviewers’ queries.
Reviewer 3
The reviewer wrote: The manuscript presents a relevant topic given the current health context. It is interesting to note the guidance provided on possible future measures to be implemented to increase literacy in the population. I also congratulate you on the effort you have made to obtain a fairly representative sample of the population. However, as in any research study, some details need to be reviewed:
Lines 81-83: Please consider transforming this sentence as the main objective of this study ("The main objective of this study is to assess...").
R: The sentence was modified according to the reviewer’s comment
Table 1: There is no need to comment on the more specific aspects of HLS-EU-Q16 (for example) if it is not mentioned in the rest of the manuscript.
R: Table 1 presents the score and cutoff levels used for the 47 items questionnaire implemented in the study, in comparison with shorter versions of the same questionnaire at 47 items; therefore we retain it would be informative to mention and describe the score and the cutoff levels of both the overall questionnaire (at 47 items) and of the shorter versions in the table.
Line 112: Consider changing "Sørensen and colleagues" to "Sørensen et al." to improve the uniformity of wording in the manuscript when referring to multiple authors.
R: The sentence was modified according to the reviewer’s comment
Lines 112, 148-152 and 390: I think there are double spaces and extra spaces in these lines.
R: Double spaces have been deleted
Lines 157, 167 and 193-194: Similar to the previous recommendation, there are spaces between signs and numbers (when on other occasions, e.g. on the same line 157, there are no such spaces). Please revise this detail throughout the manuscript to improve the uniformity of the presentation of numerical results.
R: The manuscript was revised and uniformed
Lines 124-127: Did the online questionnaire system allow for the mandatory answering of items to avoid missings? After reading these lines, it seems that it was not possible.
R: Actually, the option of a “missing answer” was possible. In this case we considered the answer “not valid”. We have specified that “If less than 80% of the items contained valid responses, the score was set to "missing".
Line 182: Please add the name of the questionnaire.
R: We added the name of the section “Corona Health Literacy Questionnaire”
Table 2: Could you transcribe the questionnaire items/questions exactly? Grammatically, they are not well written, and this would allow the study to be reproduced in other geographical and temporal contexts. Depending on the rewrite, you may need to change the heading "Question" to another label.
R: We have changed the label and rewritten some of the items.
Line 189: Southern or Southwest? On the other hand, you may consider "South and Islands", as this is the term used to refer to this geographical area. Also, if you are going to make any changes, please note that this information also appears in the abstract.
R: The sentence was modified as follows: “…, living in the ‘South and Islands’ geographical area…” (line xxx)
Lines 218 and 328, Table 5: Please change "gender" to "sex" (or "genders" to "sexes"). The sex variable is biological, which is the variable used in this study. More information is available in the section "Sex and Gender in Research" (https://www.mdpi.com/journal/ijerph/instructions#ethics).
R: We agree and have replaced gender with sex.
Line 312: Add the geographical location of Tuscany in Italy, to be able to compare with the geographical divisions you have established for this study.
R: We have added the geographical area (Central Italy)
Lines 338-342: The need to increase health literacy on coronavirus in terms of recognition of infection and symptoms should be emphasised, as these have been the most difficult tasks for respondents.
R: We agree the concept is relevant. We have modified the discussion accordingly (lines xxx): “These results highlight a specific lack of information that need to be addressed with tailored intervention, although it is important to note that the judgment on information about coronavirus symptoms indicating they had a possible coronavirus infection could be influenced by the emotional distress related to the fear of being infected, in addition to skills and competences. On the other hand, the correlation between HL-COVID, GHL-Q47 and HL-VAC suggests that general and specific skills are related, and the dimensions are, at least partly, overlapped”
Best regards,
Aldo Rosano, on behalf of the authors
Round 2
Reviewer 1 Report
All the suggestions proposed were followed by the authors, in particular the HL vaccination. Some sentences have been rephrased in the correct form. The conclusions have also been reworked in an acceptable form.